# Influence of the Metabolism on Myeloid Cell Functions in Cancers: Clinical Perspectives

**DOI:** 10.3390/cells11030554

**Published:** 2022-02-05

**Authors:** Thomas Boyer, Céline Blaye, Nicolas Larmonier, Charlotte Domblides

**Affiliations:** 1CNRS UMR5164, ImmunoConcEpT, Site de Carreire, University of Bordeaux, 146 Rue Léo Saignat, 33076 Bordeaux, France; tboyer@immuconcept.org (T.B.); cblaye@immuconcept.org (C.B.); nicolas.larmonier@u-bordeaux.fr (N.L.); 2Department of Life and Medical Sciences, University of Bordeaux, 146 Rue Léo Saignat, 33076 Bordeaux, France; 3Department of Medical Oncology, Bergonié Institute, 229 cours de l’Argonne, 33076 Bordeaux, France; 4Department of Medical Oncology, Hôpital Saint-André, 1 rue Jean Burguet, University Hospital Bordeaux, 33076 Bordeaux, France

**Keywords:** metabolism, myeloid cells, macrophages, myeloid-derived suppressive cells, cancer, immunotherapy, immunomodulation, therapeutic strategies

## Abstract

Tumor metabolism plays a crucial role in sustaining tumorigenesis. There have been increasing reports regarding the role of tumor metabolism in the control of immune cell functions, generating a potent immunosuppressive contexture that can lead to immune escape. The metabolic reprogramming of tumor cells and the immune escape are two major hallmarks of cancer, with several instances of crosstalk between them. In this paper, we review the effects of tumor metabolism on immune cells, focusing on myeloid cells due to their important role in tumorigenesis and immunosuppression from the early stages of the disease. We also discuss ways to target this specific crosstalk in cancer patients.

## 1. Introduction

The development of immunotherapies such as immune checkpoint inhibitors (ICIs) have led to impressive improvements of patient outcome. However, many patients do not respond to these therapies or eventually relapse [1]. Furthermore, immune-related adverse events, although rare, can be severe and sometimes compromise patient prognosis [2]. Therefore, there is a crucial need to fully decipher the molecular mechanisms underlying the absence of response to immune-based therapies (i.e., primary resistance) and patient relapse (i.e., secondary resistance), with the goal of identifying biomarkers and/or signatures to better stratify patients that could benefit from immunotherapy [3]. The mechanisms of resistance to immunotherapy engaged in primary and secondary resistant tumors are complex and involve the tumor, its microenvironment, and the host’s characteristics [4] including metabolic alterations, which substantially shape the nature of the tumor microenvironment (TME) and play a central role in the development of resistance to immunotherapies at all tumor stages.

Metabolic studies have focused for many years on deciphering metabolic changes occurring in cancer cells. However, the importance of metabolic changes occurring within immune cells during tumor development (and or therapies) has only been recently appreciated. Numerous studies have demonstrated that many functions of immune cells (e.g., migration and homing to a tumor site, differentiation, and anti-tumoral versus tumor-promoting activities) are regulated by or depend on the activation of specific metabolic pathways in different immune cell subsets [5,6].

With the development of ICI therapies, more attention has been paid to T lymphocytes as the primary anti-tumoral effector cells and the main targets of these immune-based therapies. However, different subsets of cells of myeloid origin can also be found in tumor microenvironments, where they may exert both anti- and protumoral impacts [7]. In this context, evidence has suggested that different subpopulations of myeloid cells participate in the complex immunosuppressive networks responsible for tumor escape from immune recognition and elimination. Interestingly, beyond their immunosuppressive functions, these myeloid cells can also exert direct or indirect non-immunological protumoral functions, such as the promotion of tumor cell growth, survival, invasive properties, and angiogenesis, and they can foster many steps of the metastatic process. This current review focuses on the analysis of the main metabolic pathways involved in the regulations of two myeloid cell subpopulations primarily constituting the myeloid landscape within the tumor microenvironment, namely myeloid-derived suppressor cells (MDSCs) and tumor-associated macrophages (TAMs). We further discuss strategies for targeting these metabolic pathways in MDSCs and TAMs in light of their therapeutic development in clinical settings.

## 2. Metabolism of TAMs and MDSCs, the Major Components of the Tumor Myeloid Landscape

Functionally, myeloid cells constitute essential elements of the immunoinhibitory networks responsible for tumor escape from anticancer immunity. They therefore represent major impediments for immune-based interventions. However, beyond their cardinal immunosuppressive functions, myeloid cells are also equipped with a broad array of “non-immunological” tumor-promoting functions. Indeed, accumulating evidence has demonstrated that these cells can directly promote primary tumor cell survival and proliferation by fostering tumor neoangiogenesis and invasion [8]. They are also capable of remodeling the extracellular matrix and participating in local tissue invasion [9]. Moreover, they participate in the pre-metastatic niche formation before the arrival of cancer cells, thus contributing to the preparation of the “soil” for seeding by metastatic tumors [10,11]. The role of cancer-induced myeloid cells in resistance to chemotherapy, radiotherapy, and endocrine therapy has also been described, thus making them potential targets for the development of new immunotherapies.

In this review, we focus on two myeloid subtypes, macrophages and MDSCs, which are key players of immunosuppression in the tumor microenvironment. Biological features have been extensively reviewed elsewhere [12,13]. Macrophages represent an innate immune cell population encompassing a large spectrum of phenotypes, depending on environmental cues. They are endowed with a high degree of plasticity, exhibiting dedicated functional programs in response to specific environmental signals [14]. Indeed, new technologies as scRNA-Seq may help in identifying the different immune subpopulations and their dynamic functional switch due to transcriptional reprogramming, which leads to important plasticity [15]. Two extreme polarization states were originally described, but in practice, many different intermediary states exist as a continuum between antitumoral M1 and protumoral M2 that are difficult to clearly delineate in vivo [16]. The shift between tumor-promoting and tumor-suppressing macrophages in cancer has been subjected to extensive work aimed at defining anti-cancer strategies to functionally reprogram protumoral into antitumoral macrophages. On the other hand, MDSCs have traditionally been described as morphologically, phenotypically, and functionally heterogeneous subpopulations of myeloid cells blocked at early differentiation stages. Historically described in the context of malignancies, the essential role played by these cells has been extended to other physiological and pathological areas.

Notably, evidence has suggested that selected metabolic pathways can control myeloid cell recruitment, differentiation, and function. Furthermore, the heterogeneity of myeloid cell polarization is correlated to the balance between different metabolic pathways depending on tumor metabolism. Glucose is the most important molecule that controls immune cell differentiation and functions under the control of hypoxia through HIF-1α and mTOR [17]. Indeed, an increase in the glycolytic rate of immature myeloid cells under the control of hypoxia favors their differentiation into MDSCs and their expansion, with a decreased oxidative metabolism [18]. Glycolysis generates phosphoenolpyruvate, which is involved in MDSC survival, protecting them from reactive oxygen species (ROS) toxicity [19]. MDSCs also rely on fatty acid oxidation (FAO) to support their immunosuppressive functions [20]. However, the absence of the consensual characterization of the different subtypes of MDSCs makes the analysis of their metabolism complex, as several reports have highlighted that different subpopulations of MDSCs may harbor different—even opposite—metabolic requirements [21]. Macrophages also strongly rely on glucose catabolism. Indeed, they fuel their energy requirements with enhanced glycolysis for the M1 inflammatory phenotype, whereas they switch toward an oxidative phenotype, favoring OXPHOS and FAO for the M2 immunosuppressive phenotype [22,23]. In the tumor microenvironment, this opposition between the M1 and M2 phenotypes is not as clear, and both phenotypes may participate in tumor progression. Indeed, inflammatory macrophages are involved in the secretion of proangiogenic and proinflammatory factors through HIF-1α expression, favoring tumor progression [24,25]. Furthermore, HIF-1α expression also induces iNOS activation, with an arginine consumption for NO production [26]. Under normal conditions, NO is strongly involved in host’s defense against pathogens, but in cancer, it favors genetic instability. Finally, in these myeloid cells, the shift toward glycolysis induces the accumulation of succinate due to a truncated tricarboxylic acid cycle that sustains aerobic glycolysis through HIF-1α expression. Furthermore, increased glycolysis induces a decrease in mitochondrial respiration that leads to the accumulation of ROS in the inflammatory phenotype. Finally, GCN2 (general control nonderepressible 2) is a kinase activated by amino acid and glucose deprivation that controls cellular metabolism and functions. Recently, GCN2 was reported to regulate macrophage and MDSC polarization and functions, highlighting its key role in the immunosuppressive microenvironment [27].

Highly glycolytic tumor and myeloid cells release high amounts of lactic acid in the immunosuppressive microenvironment, thus leading to acidification. Lactate is involved in the recruitment of myeloid cells within tumors [28]. Furthermore, it induces the expression of proangiogenic factors (such as vascular endothelial growth factor A (VEGF-A)) and arginase-1 in the immunosuppressive phenotype of macrophages and MDSCs [29,30]. 

Glutamine is another important metabolite involved in glutaminolysis that strongly supports functions of immunosuppressive macrophages and MDSCs. Indeed, along with tumor cells, macrophages and MDSCs are some of the most important glutamine-consuming cells because they express high levels of glutaminase [31]. Glutaminolysis is essential in myeloid cells to feed the TCA cycle, sustaining the production of inflammatory cytokines in macrophages and upregulating M2 markers such as arginase-1 via epigenetic regulations [32]. In MDSCs, glutamine consumption favors the expansion of MDSC pools within tumors [33].

After their differentiation and maturation, myeloid cells also rely on metabolism to exert their protumoral inhibitory functions. The most important pathways are arginine, cysteine, and tryptophan amino acid metabolism; the adenosine pathway; and ROS production. In this review, we discuss the main metabolic pathways involved in cell differentiation and maturation, the principal metabolites involved in tumor-promoting myeloid cell functions, and their potential therapeutic targeting pathways in human cancer disease. Clinical trial data are shown in Table 1. Though metabolites are described elsewhere, metabolic pathways are interconnected within tumor and immune cells, meaning that targeting one pathway could favor metabolic reprogramming in the tumor microenvironment, thus inducing different resistance mechanisms. 

## 3. Aerobic Glycolysis, Lactate Accumulation and Acidification of the Medium

Glucose represents a primary source of energy at the cell level. Its breakdown by glycolysis leads to pyruvate, which enters into the mitochondria and is transformed by oxidative decarboxylation into CO_2_ and redox cofactors. These cofactors are then used for oxidative phosphorylation (OXPHOS) to produce ATP (36 moles). However, under hypoxic conditions, pyruvate remains in the cytoplasm and is converted into lactate in a process called fermentation, leading to the lower production of energy via ATP (only 2 moles). AMP-activated protein kinase (AMPK) and mammalian target of rapamycin (mTOR) are master regulators of cell metabolism and specifically of glycolysis, but they are associated with opposite functions: AMPK is involved in catabolism and mTOR is involved in anabolism. AMPK is an upstream negative regulator of mTOR activation through interactions with Raptor, which favors OXPHOS increases [34]. 

Cancer cells preferentially use aerobic glycolysis and lactate fermentation under normoxic conditions to produce intermediates and precursors essential for cell proliferation: this is the Warburg effect [35]. Because aerobic glycolysis is less efficient than OXPHOS at producing energy, tumor cells increase their glucose uptake and overexpress glycolytic enzymes to sustain their high proliferative rate [36]. Indeed, cancer cells have a high glucose consumption rate that leads to a decrease in the extracellular pool of glucose. The Warburg effect generates large amounts of lactic acid, which is released into and consequently acidifies the TME [37].

Glucose metabolism favors an immunosuppressive microenvironment. As discussed above, tumor and immunosuppressive myeloid cells induce glucose deprivation in the TME due to their high glycolytic rates. However, glucose is also required for the activation and function of T cells, and glucose deprivation induces global T cell anergy and a reduced antitumor immune response [38,39]. Furthermore, myeloid cells must be highly glycolytic to sustain their differentiation and functions and they participate in glucose deprivation, as discussed previously [22]. Most of the glucose is used in aerobic glycolysis, leading to the release of large amounts of lactate of medium acidification, leading to T cell anergy in human and mouse models, with a blockade of their proliferation and functions [40]. Furthermore, lactic acid increases the differentiation and infiltration of MDSCs within the TME, thus sustaining their immunosuppressive functions, and it polarizes TAMs into the M2 phenotype with strong immunosuppressive effects on T lymphocytes, inducing the expression of arginase I and iNOS [28,30].

Targeting aerobic glycolysis and TME acidification is challenging due to the involvement of this metabolic pathway in a plethora of biological processes, though it may be achieved by focusing on glucose uptake by tumor cells to increase glucose availability, inhibit glycolytic enzymes or lactate-producing enzymes, or block lactate efflux. The aim of these treatments is to decrease the differentiation and immunosuppressive functions of MDSCs and TAMs within the TME. The inhibition of glucose uptake from the TME has been proposed as an interesting strategy to increase the availability of glucose in the TME while inhibiting myeloid cell differentiation and functions. Several agents are currently being developed in the preclinical setting, with interesting results. However, to date, only silybin, a flavonoid inhibiting the glucose transporters (GLUTs) by direct interaction with GLUT4, has been evaluated in very few clinical trials on small number of patients (Figure 1). In prostate cancer, silybin was found to be associated with clinically manageable liver toxicities, but no biological responses were observed [41]. There are currently no ongoing trials with this molecule, and other small molecules that inhibit GLUT are currently being assessed in clinical trials. 

The inhibition of glycolytic enzymes such as hexokinase has been proposed as a more promising approach. Hexokinases are involved in the first step of glycolysis. Hexokinase-2 (HK2) is overexpressed in cancer and has been associated with relapses and poor outcomes in solid tumors [42]. HK2 could be targeted by 2-deoxyglucose (2DG), which is a competitive analog of glucose (Figure 1). 2DG was found to induce metabolic reprogramming from glycolytic to OXPHOS metabolism, reducing lactate production and limiting the aggressiveness of the disease in vivo [43]. When combined with docetaxel in advanced prostate cancer, it was associated with a limited efficacy, and tolerance was marked by hyperglycemia observed in the whole cohort, which is an important issue in its further development (NCT00633087, unpublished results). Other hexokinase inhibitors have been developed, but no trials are ongoing in humans. 

Metformin and phenformin, biguanides used in antidiabetic therapies, are able to selectively inhibit HK1 and HK2 and to activate AMPK (Figure 1). Several clinical trials have explored the potential of metformin, alone or in combination with other drugs, to promote antitumor immune responses in several types of human cancers. However, there have been no reported results on the modulation of myeloid subpopulations by these molecules in clinical trials. In vitro and in vivo evidence has suggested that these molecules are able to reprogram T lymphocytes towards AMPK-driven metabolisms, protecting CD8+ T cells from apoptosis [44]; they also induce the degradation of PD-L1, with a diminution in T cell exhaustion [45]. On myeloid cells, phenformin has been shown to inhibit MDSCs in in vivo and ex vivo melanoma models [46]. Furthermore, metformin was found to lead to a decrease in the accumulation and immunosuppressive functions of MDSCs, thus leading to an increase in the tumor infiltration of CD8+ T cells in mice models [47,48]. 

The specific direct targeting of the metabolic master regulator mTOR is also under evaluation in several clinical trials, with the leader molecule being rapamycin (Figure 1). mTOR inhibition prevents TAM differentiation into M2 macrophages and promotes the activation of M1 macrophages [49]. These observations are not surprising because M2 macrophage polarization and functions rely on mTOR, as discussed above [22]. Furthermore, mTOR inhibition has been reported to impair MDSC expansion, accumulation into the TME, and suppressive functions [50]. In humans, the metabolic effect of mTOR inhibition has not aroused specific interest, and its inhibition is clinically challenging insofar as mTOR is critically involved in multiple physiological processes such as the control of immune cell differentiation and functions. 

In this context, it is noteworthy that immune checkpoints inhibitors may also exert impacts on immunometabolism. Indeed, PD-L1 expression on tumor cells induces the activation of the Akt/mTOR pathway, leading to high rates of glycolysis [51]. A PD-L1 blockade using ICI induces a decrease in glucose consumption and an increase in glucose availability within the TME [52]. M2 macrophages could upregulate PD-L1 at their surface [53]. On T cells, PD-1 engagement induces a decrease in glycolysis, thus leading to T cell anergy, and its blockade activates glycolysis and T cell effector functions [53] in parallel with an increase in the extracellular pool of glucose [54]. Likewise, blocking MDSC PD-1/PD-L1 expression induced by HIF-1α (under hypoxia conditions) leads to the inhibition of their immunosuppressive functions by downregulating IL-6 and IL-10 secretion [55]. 

An additional strategy may consist of reducing the acidification of the environment by inhibiting lactate dehydrogenase A (LDHA) and/or the monocarboxylate transporters (MCTs). Several small inhibitors specific to LDHA and/or LDHB have been developed, but none of them have been assessed in clinical trials. The main strategy available to block extracellular acidification in the context of malignancies consists of the inhibition of MCT transporters, responsible for the efflux of lactate from tumor cells. MCT transporters are used in lactate influx/efflux to avoid the acidification of intracellular cytoplasm of cancer cells. The blockade of MCT leads to the accumulation of lactates into its cytoplasm, inhibiting glycolysis and inducing the accumulation of ROS [56]. AZD3965 has been developed as a potent inhibitor of MCT1 with in vivo inhibitory activity in a breast cancer mouse model [57] (Figure 1). This molecule has been assessed in metastatic solid tumors (NCT01791595), but the results have not been reported. In lymphoma, ADZ3965 showed a good safety profile but a moderate Disease Control Rate (DCR) of 11% when used alone [58]. It is noteworthy that cases of “hyper-Warburgism” have been reported due to the high release of lactate by tumors being increased by AZD3965 and leading to malignant lactic acidosis. This has resulted in the suspension of trial recruitment and the exclusion of patients with high blood lactate levels [59].

One limitation of targeting glucose metabolism is the broad expression of transporters and enzymes by non-malignant cells, as well as the off-target effects of most inhibitors, leading to potential severe side effects such as lactic acidosis and hyperglycemia. Furthermore, the blood–brain barrier expresses high levels of GLUT1, and its inhibition could lead to severe neurologic toxicities. Another issue is the lack of transferability between human and mouse models, with some glucose transporters expressed in humans but not in mice, underlining the limit of mouse models to accurately predict toxicities in humans. To conclude, glucose targeting is an interesting approach with important issues that can impede its development in humans, and it is important to assess its specific effects on myeloid cells because it has not been further described in the clinical setting.

## 4. The Amino Acid Metabolism

### 4.1. Glutamine and Glutaminolysis

Glutamine is the most abundant amino acid in circulation. High glutamine consumption has been detected in multiple cancer types. Glutamine enters in the tricarboxylic cycle in mitochondria to generate several intermediates involved in cellular processes. Glutamine is essential for the growth of most tumors because it provides carbon and nitrogen essential for many metabolic cascades. This glutaminolysis mainly depends on the expression of the glutaminase (GLS) enzyme. 

In case of glucose deprivation, glutamine metabolism is crucial because glutamine can be used as an alternative pathway for energy supply. Indeed, cells can adapt their metabolism according to nutrient availability, especially in the tumor microenvironment, where MDSCs and tumor cells are the main glutamine consumers. The high consumption rate of glutamine by cancer cells exacerbates the metabolic competition between tumors and the immune system. Glutamine deprivation leads to T lymphocyte exhaustion and contributes to their repolarization toward a regulatory (tumor-promoting) phenotype [60].

Glutamine participates in the generation of MDSCs from immature myeloid cells, and glutamine metabolism provides intermediates for MDSC differentiation and recruitment [18]. Furthermore, glutamine oxidation is involved in the control of the immunosuppressive functions of these tumor-promoting myeloid cells. Macrophages also express high levels of GLS, as glutamine is essential for their functions, such as cytokine production, antigen presentation, phagocytosis, and protumor activities.

Promising clinical studies attempting to modulate glutamine metabolism by using glutamine analogs (DON (6-diazo-5-oxo-L-norleucine); acivicin), targeting multiple enzymatic steps of glutamine metabolism including GLS, have been conducted (Figure 2). However, all of them had to be suspended due to important toxicity and side effects in patients, specifically severe gastrointestinal toxicities due to local glutamine deprivation [61]. These disappointing results are mainly due to the fact that targeting the glutamine pathways using analogs is not specific enough and includes a broad spectrum of multiple pathways on both cancer and not-malignant cells. Of note, a recent murine study showed that the deprivation of glutamine in the tumor microenvironment led to an increased expression of G-CSF and GM-CSF in a mouse mammary model, resulting in the generation of MDSCs [62]. These data again suggest an important need to develop more specific drugs targeting part of the glutamine metabolic pathway rather than a non-specific global deprivation of this essential amino acid. The pharmacological optimization of targeted therapies could improve their efficacy while limiting toxicities. Indeed, encouraging studies have emerged on the use of glutamine pathway inhibitors in mice that would be more specific and seem to significantly restrict tumor growth. J. D. Powell and collaborators recently reported that the use of JHU-083, a glutamine antagonist prodrug of DON, suppressed tumor growth in a mice model of triple-negative breast cancer. The administration of this molecule reduced the level of G-CSF (CSF3) produced by tumor cells, with (as a consequence an impaired recruitment of MDSCs) increased immunogenic cell death and the promotion of an inflammatory environment with increase in M1-like TAMs potentially arising from the differentiation of MDSCs [33]. This effect was observed in not only the TME but also the metastatic sites. Furthermore, an increased infiltration of CD8+ cytotoxic T lymphocytes in the tumor site was detected and associated with restored antitumor immune functions and improved survival [63]. In addition, JHU-083 was observed to overcome tumor resistance to immunotherapy. Of note, another prodrug, DRP-104 (Sirpiglenastat) (Figure 2), is currently been evaluated in a clinical trial in combination with atezolizumab (NCT04471415). 

Glutaminolysis could be more inhibited by specifically targeting GLS due to its central role in the consumption of glutamine. Indeed, GLS is expressed by a wide spectrum of tumors, and its overexpression is strongly correlated to patient outcomes and a more aggressive disease [64]. The inhibition of GLS enzymes has shown promising results, with direct antitumor effects [65]. Importantly GLS inhibition also leads to indirect antitumor effects such as the restoration of antitumor T lymphocyte functions [66]. Among those selective inhibitors, CB-839 (telaglenastat) seems to be the most promising (Figure 2). Several ongoing studies are currently assessing its efficacy in different tumors, alone or in combination with PARP inhibitors, chemotherapies, or tyrosine kinase inhibitors. Only two trials with CB-839 have reported results in metastatic renal cell carcinoma: the CANTATA (NCT03428217) and ENTRATA (NCT03163667) trials, which assessed CB-839 with cabozantinib (a VEGF receptor inhibitor) and everolimus (a mTOR inhibitor), respectively. In the CANTATA trial, no statistical differences were observed between patients receiving cabozantinib alone or in combination with CB-839. However, objective response rates (ORRs) were higher in the subgroup of patients pretreated with an anti-PD-1 therapy (32%) or with an anti-PD-1/anti-CTLA-4 combination (37%) compared to non-pretreated patients (20%), with a two-month increase in the median progression-free survivals (mPFS) [67]. This emphasizes that metabolism targeting could overcome therapeutic resistance to immunotherapy, and sequential or combinatorial strategies need to be precisely assessed.

Another small GLS inhibitor currently being assessed, IACS-6274 (IPN60090), leads to the sustained inhibition of GLS (Figure 2). Results from the first phase I trial were presented at the international congress ASCO in 2021 [68]. Biologically, a decrease in the glutamate/glutamine ratio was observed in PBMC samples as early as after the second week of treatment, with a decrease in GLS activity of 80%. The molecule demonstrated an interesting efficacy profile, with a DCR of 60% at 3 months and 30% of patients presenting a sustained response longer than 6 months. However, due to the small number of patients, specific data of patients with immunotherapy resistance were not assessable. These results are encouraging, as stabilizations were observed in heavily pretreated patients (all of the cohort received at least two previous lines of treatment, and 50% received at least five lines). This study focused on molecularly-selected patients, but data were not reported, and IACS-6274 must be assessed in combined therapy to increase its efficacy. Based on these observations, and because the in vitro and in vivo inhibition of glutamine metabolism is associated with the upregulation of PD-L1 expression by cancer cells, there is a strong rationale to combine glutamine metabolism inhibition with immunotherapy in immune-sensitive or immune-resistant tumors, as discussed above [69].

As a recurrent issue with metabolism targeting, the selection of patients who may selectively benefit from glutamine metabolism targeting is critical. Further complicating the situation, the metabolic pathways used by cancers are different depending on tumor origins, and the microenvironment composition of myeloid cells can differ. Furthermore, because of their high degree of plasticity, cancer and immune cells can reprogram their metabolism to adapt to changing environment conditions or therapies. There is therefore an important need to identify glutamine-addicted tumors with either the pathological analysis of tumor samples (the overexpression of glutamine pathway enzymes by immunohistochemistry, for example) or imagery. For instance, gliomas are tumors that strongly consume glutamine. Accordingly, the possibility of identifying highly affine tumors for glutamine by imagery such as PET has been studied [70]. However, the specific involvement of glutamine-consuming myeloid cells has not been assessed in these studies even though glutamine is an important metabolite involved in the polarization and immunosuppressive functions of myeloid cells. The myeloid infiltration within tumors can be an interesting predictive biomarker, which is easily assessable through immunohistochemistry, under the hypothesis that tumors with high infiltration rates of glutamine-consuming myeloid cells may harbor high response rates under glutamine-targeted therapies.

### 4.2. Arginine Depletion

One promising therapeutic approach extensively studied in clinical oncology is the targeting of arginine deprivation. Arginine is a semi-essential amino acid involved in the urea cycle and the biosynthesis of nitric oxide (NO), polyamines, and glutamate. Arginine can be sourced from the extracellular pool or produced in the urea cycle through the transformation of citrulline into arginine by the argininosuccinate synthetase (ASS) and argininosuccinate lyase (ASL) enzymes, which are rate-limiting enzymes. 

Arginine is central for multiple essential cellular functions such as cell proliferation, survival, and protein synthesis. One important characteristic of tumor cells is their auxotrophic state for arginine due to defects into the biosynthesis enzyme machinery, specifically ASS, ASL, and ornithine transcarbamylase (OTC) [71]. Due to important requirements in arginine, malignant cells take arginine from the extracellular pool. As a consequence, arginine availability for healthy epithelial cells and immune cells is reduced, leading to their dysfunction. Therefore, specifically targeting cancer cells with arginine-depriving strategies could be of potential interest insofar as it would spare healthy tissues that harbor preserved ASS and ASL enzyme activities while impairing the cell growth and proliferation of cancer cells, eventually leading to cell death [72]. 

Importantly, there are different major enzymatic compounds able to catabolize free arginine. Arginase is one of these enzymes, and it has been shown to be overexpressed in some cancer types and sustain tumor cell proliferation, with a negative impact on patient outcomes [73]. Furthermore, MDSCs and TAMs highly express arginase enzyme, specifically arginase 1 (Arg1) [74], which substantially contributes to the immunosuppressive function of MDSCs and TAMs [75]. MDSCs can also release granules containing Arg1 into the extracellular matrix, increasing the depletion of L-arginine and immunosuppression [76]. Arginine depletion mediates immunosuppression through the inhibition of T-cell proliferation (inhibition of TCR expression), decreases inflammatory cytokine production [77], and induces regulatory T (Treg) polarization in cancer patients [78].

Arginine is also depleted by iNOS (inducible nitric oxide synthase) through its transformation into urea and L-ornithine. iNOS is expressed by both macrophages and MDSCs. This induces the production of NO, which reacts with superoxide and produces reactive nitrogen intermediates (RNS). NO and RNS are involved in the inhibition of T cell proliferation and exert a direct apoptotic effect on T lymphocytes [79]. Furthermore, increase in NO cytosolic levels induces the generation of ROS in MDSCs and inhibits the downstream signaling of the IL-2 receptor. RNS also favors the nitration of chemokine, leading to decreased intratumor T cell infiltration [80]. Finally, NO production is involved in recruitment of MDSCs, TAMs, and Tregs. iNOS should be targeted using tadalafil (a PDE-5 inhibitor) or NAC, which reduce iNOS expression with MDSCs and increases the efficacy of ICI or adoptive cell transfer, respectively [81,82]. Furthermore, targeted therapies aiming to decrease RNS levels have produced limited results in humans. Indeed, a better understanding of iNOS’s role in immunosuppression has highlighted the redundancy between the different metabolic pathways (ARG1 and iNOS), meaning that targeting one should not be efficient enough and combinatory strategies need to be developed. 

In this context, targeting the immunosuppressive tumor and myeloid cell-mediated depletion of arginine is of great interest in the clinical setting to restore lymphocyte activation by shifting the balance in favor of the accumulation of arginine. Indeed, murine studies have shown that the inhibition of the activity of arginase 1/2 effectively decreases the growth of tumors [77,83]. Furthermore, the use of INCB001158 (a specific inhibitor of this enzyme) was associated with a diminution in the myeloid cell-mediated immunosuppression on T lymphocytes in an in vitro model [84] (Figure 2); its administration alone or associated with chemotherapy, adoptive cell transfers, or ICI induced an increased infiltration of CD8+ and NK cells within tumors, with a restoration of the secretion of proinflammatory cytokines, thus leading to an inhibition of tumor growth in murine cancer models [84]. These observations indicate that combining metabolic targeting therapies with currently used ICI may represent an interesting strategy to reverse resistance.

Different molecules are currently being developed in cancer as monotherapies or in combination with chemotherapy or immunotherapy. However, most relevant studies are focused on lymphoid subpopulations, and few data regarding the effect of arginine targeting on myeloid cells are available. These molecules are classically able to inhibit intracellular and extracellular Arg1, such as the enzyme protected in extracellular vesicles. In the clinical setting, INCB001158 has mostly been studied in combination with immunotherapy targeting the PD-1/PD-L1 axis or with chemotherapy. However, one of the main issues with Arg1 inhibitors may be the inhibition of the urea cycle into liver cells involved in detoxication of ammonia, leading to a theoretical increased risk of hyperammonemia. However, a good safety profile and no sign of hyperammonemia have been reported after the administration of INCB001158 in mouse models and clinical trials [84]. In terms of efficacy, INCB001158 showed promising results in combination with pembrolizumab in colorectal cancer with stable microsatellite (MSS), with a DCR of 37% compared to 28% with INCB001158 alone [85]. The ORRs were 3% with INCB001158 alone and 6% when combined with pembrolizumab. These data have to be compared with results obtained with pembrolizumab alone in MSS colorectal cancer, with a DCR of 17% and an ORR of 0%, highlighting the clinical benefit of combining ICI with metabolic targeting therapies [86]. Another phase I trial assessed the combination of INCB001158 with chemotherapy in advanced/metastatic solid tumors (NCT03314935); this association resulted in interesting efficacy in advanced biliary tract cancers, with an ORR of 24%, a DCR of 67%, and an mPFS of 8.5 months [87]. In the pivotal chemotherapy trial in the same clinical setting (advanced/metastatic biliary tract cancers), INCB001158’s efficacy appeared modest as a first-line result. Indeed, the association of cisplatin and gemcitabine as first-line treatments resulted in a DCR of 81.4% and an mPFS of 8.0 months [88]. Further investigations have to be conducted, specifically on therapeutic associations.

The arginine pathway could also be targeted through an arginine depletion strategy using ADI-PEG or rhARG-PEG (Figure 2). These molecules correspond to the ADI or Arg1 enzymes conjugated to polyethylene glycol (delivered in order to consume extracellular arginine), favoring its depletion into the tumor microenvironment. Though these molecules harbor interesting antitumor efficacies due to the auxotrophic state of cancer, a deleterious effect on effector T cell responses has been reported in vivo through the induction of MDSCs, highlighting the need of combined therapies targeting MDSCs [89]. 

One interesting strategy could be to increase T cell immunity against ARG1-expressing cells, such as tumor and myeloid cells. Indeed, in vivo models have shown increases in antitumor immunity within tumors and, when synergized with anti-PD-1 therapy, reduction in the immunosuppressive functions of myeloid cells, favoring a shift toward an antitumor phenotype of macrophages [90]. There are no currently ongoing clinical trials using this strategy.

Finally, innovative therapies should be developed to increase the efficacy of immunotherapies. One such innovative treatment is the use of engineered probiotics able to convert ammonia into L-arginine within tumor microenvironments. Preclinical data have demonstrated a synergistic effect with anti-PD-1 molecules [91]. Another interesting strategy may be the inhibition of ARG1 synthesis using a transcriptional inhibitor such as AT38 [80], which was altered in a preclinical model of the pancreatic cancer immunosuppressive functions of M2 and MDSCs. However, there are no ongoing clinical trials assessing these strategies [92].

The arginase inhibitors that are currently developed are competitive inhibitors of both Arg1 and Arg2 due to the strong homology of their sequences in the catalytic sites, which are the target of these inhibitors. However, their tissue distributions are different, and targeting specifically one of both isoforms is challenging but could be interesting insofar as it would give better insights into the role of each isoform [93].

An improved targeting of Arg1 requires the better identification of which patients may respond to these therapies. This may be achieved by identifying predictive biomarkers such as the kinetics of arginine modifications in the blood or the level of overexpression of Arg1 into the tumor tissue and/or myeloid subpopulations. Thanks to immunochemistry approaches, arginase expression is easy to detect, making it a rapidly identifiable biomarker for the use of arginase inhibitors as immunomodulatory agents. However, the cut-off for arginase expression to be considered overexpressed in tumor still needs to be defined. Furthermore, the distinction between arginase inhibitors’ impact on tumor cells versus on myeloid cells remains difficult to ascertain. Finally, most clinical trials have assessed the effects of arginase inhibitors concomitantly combined with other therapies such as chemotherapies and/or immunotherapies. However, a sequential strategy consisting of first reverting the immunosuppressive TME using an arginase inhibitor and then activating antitumor immune responses using anti-PD-1/PD-L1 may be more efficient. This question will be partially answered in the NCT02903914 clinical trial that is assessing the effect of this arginase inhibitor in patients who were previously treated with immune checkpoints inhibitors.

### 4.3. Tryptophan–Kynurenine Pathway

Tryptophan is an essential amino acid that is involved in the promotion of malignant cell development and the mechanisms of immunosuppression in cancer-bearing hosts. Its oxidative catabolism is regulated by three major enzymes: two isoforms of indoleamine-pyrole 2,3 dioxygenase (IDO1 and IDO2) and tryptophan 2,3 dioxygenase (TDO), which degrades tryptophane into kynurenine. These enzymes catalyze the rate-limiting step of tryptophan metabolism [94,95]. While IDO2 and TDO are expressed by a wide diversity of cells, IDO1 is highly expressed by stromal and myeloid immune cells (including TAMs and MDSCs) and its expression is triggered by inflammatory stimuli [96]. Furthermore, IDO1 expression is linked to ARG1 expression in dendritic cells, as these cells produce arginine-derived molecules involved in the phosphorylation of IDO1 and its long-term activity [97]. IDO1 expression is augmented in diverse types of cancers and often correlates with poor patient prognosis [98]. IDO1 overexpression within tumor beds leads to tryptophan deprivation, which impairs antitumor immune functions [99]. It is noteworthy that IDO1 overexpression is also associated with resistance to ICI [100]. Finally, IDO1 is not only an enzyme but also plays a role as signal-transducing substrate responsible for DC’s regulatory phenotype in a TGF-dependent manner [101]. 

Tryptophan catabolism has major impact on T lymphocyte effector functions. Indeed, kynurenine, a byproduct of tryptophane catabolism, is involved in a lack of T cell activation [102]. Interestingly, it has been shown that IDO-mediated tryptophan degradation, by generating kynurenine, is able to induce tumor-promoting immunosuppressive Treg polarization of CD4^+^ T lymphocytes in vitro [99,103], a result in accordance with previously reported clinical data that IDO1 expression correlates with high FOXP3^+^ Treg expression in cancer patients [104,105]. Of note, preclinical studies have shown that the depletion of tryptophan inhibits T-cell proliferation and increases T-cell apoptosis, blunting antitumor responses [106]. Along these lines, a tryptophan metabolite, 3-HAA (3-hydroxyanthranilic acid) has been observed to suppress NO secretion via macrophages, thereby decreasing their ability to eliminate tumor cells [107]. Finally, high levels of IDO1 induce IL-6 expression by MDSCs in an autocrine manner, leading to the recruitment of MDSCs at the tumor site [108].

Based on the importance of tryptophan metabolism in the promotion of cancer development, clinical evaluations of agents capable of interfering with these pathways have been carried out. Interestingly, only small-molecule inhibitors targeting IDO1, not IDO2 nor TDO, have reached clinical trials. Epacadostat (INCB024360) is one such agent (Figure 2). It is an orally available, reversible competitive IDO1 inhibitor [109]. This drug showed potent anti-IDO1 activity in vitro and high selectivity for IDO1 (comparing with IDO2 and TDO). Moreover, epacadostat is efficient at increasing NK cell, effector T cell, and CD86 high dendritic cell functions, as well as blocking immunosuppressive Treg, in vitro [110]. Interestingly, epacadostat suppresses tumor growth in immunocompetent but not immunocompromised mice, highlighting its specific effect on antitumor immune response [111]. Finally, it can reverse resistance to anticancer therapies, as demonstrated in an LLC mouse model that demonstrated an increase in IDO expression in MDSCs after hypofractionated radiotherapy that was reversed using epacadostat [112]. Phase I and II trials have demonstrated that epacadostat enhances the response to anti-PD1 immunotherapy, but negative data from KEYNOTE-252/ECHO-301 phase III trial in unresectable or metastatic melanoma have stopped other IDO1-targeting clinical trials that may have shown potentially different and interesting outcomes [113,114]. New interest is currently emerging in the targeting of tryptophan metabolism in clinics with different approaches using IDO1 inhibitors (navoximod (NLG-919/GDC919), BMS-986205 (F001287), KHK2455, LY3381916, and MK-7162) or inhibitors targeting both IDO1 and TDO, all of which are currently being clinically evaluated (Figure 2). Furthermore, though the KEYNOTE-252 trial was negative, some preclinical data have highlighted a synergistic role of the combination of IDO1 inhibitors and immune-modulating therapies or chemotherapies [115,116]. Based on these data, several combinatory strategies are currently being assessed in the clinical settings by combining IDO1 inhibitors with antitumor vaccines, immunotherapies, chemotherapies, and radiotherapies [117]. Another promising strategy is to favor the activation of IDO and PD-L1-specific T cells using vaccination. This strategy was used in a phase 1/2 clinical trial in immunotherapy-naïve metastatic melanoma patients that showed good results, with an ORR of 80% and an mPFS of 26 months [118]. 

The disappointing results of IDO targeting trials may be explained by several reasons. The first is that the selection criteria between phase I and phase III have not been identical, with potential selection biases. Indeed, for phase I trials, patients have to be highly fit to be included with non-rapidly growing diseases, whereas in phase III, the inclusion of patients depends on less selective criteria. Furthermore, the translation of results from non-randomized phase I trials of a small number of patients into large phase III trials is contentious, especially when the experimental drug is assessed in combination with a validated therapy that is effective as a single molecule. 

Another issue relies on the need to better select patients that may benefit from immune-targeting and metabolism-targeting therapies. To address this issue, many studies have focused on the identification of predictive biomarkers, especially for IDO inhibition. One such proposed biomarker may be the expression levels of IDO within tumor and immune cells. However, markers to identify macrophages and MDSCs are not specific due to the plasticity of these cells. Furthermore, like other biomarkers, standardizations of the method for its assessment, the definition of the positive cut-off, and the nature of the cells (tumor and/or immune cells) in which it should be detected still need to be determined. Circulating biomarkers such as kynurenine and/or tryptophan at baseline and during treatment may also be of interest. 

In addition, the inhibition of IDO1 leads to an increase in tryptophan availability, which can then be metabolized by other enzymes and result in the production of higher levels of the inhibitory molecule kynurenine, which is endowed with high immunosuppressive functions. Tumor cells are also capable of overcoming IDO1 inhibition by overexpressing other enzymes involved in tryptophan metabolism such as TDO and IDO2. It may therefore be advantageous to target more downstream pathways. In this context, the anti-tumoral consequences of inhibiting the aryl-hydrocarbon receptor (AhR), downstream of the IDO/TDO pathway, is currently being considered in pre-clinical studies. Indeed, in a mouse melanoma model, treatment with CH-223191 alone or combined with PD-1 blockade led to an inhibition of tumor growth, with a restoration of antitumor immunity and an inhibition of Treg and immunosuppressive macrophages [119]. Other molecules, BAY2416964 and IK-175, are currently being assessed in phase I/II trials.

Of note, in this clinical evaluation, the impact of tryptophan metabolism targeting on TAMs or MDSCs has not been clearly deciphered. This is an important question considering the tumor-promoting capabilities of these immunosuppressive myeloid cells.

### 4.4. Cysteine Depletion

Another amino acid that may be of clinical relevance for cancer therapies in relation to the myeloid tumor landscape is cysteine. This amino acid greatly contributes to cancer cell survival and proliferation, and it is also implicated in the suppression of T lymphocytes by myeloid cells. 

In cancer cells, cysteine is essential in controlling oxidative stress as a precursor of glutathione and contributes to cellular bioenergetics through the production of hydrogen sulphide (H2S) [120]. Cysteine is also a major source of carbon for biomass and energy production through its conversion into glutamate, pyruvate, and coenzyme-A, which feed the TCA cycle. Indeed, cysteine metabolism is an important pathway that is interconnected with other strong metabolic pathways in cancer. Moreover, cysteine is an important determinant of treatment resistance, especially for oxidative or alkylating drugs, as cysteine and glutathione are major scavengers of free radicals (including ROS). 

T lymphocytes use extracellular cysteine to fulfill their requirements. In T lymphocytes, cysteine levels are low, mostly depending on its biosynthesis or antigen-presenting cells (APCs). Indeed, they are able to import cystine through the xc- antiporter (also called cystine/glutamate antiporter), transforming it into cysteine and releasing it into nearby T cells through the ASC transporters [121]. Interestingly, MDSCs express the xc- antiporter and are thus capable of importing cystine to catabolize it into cysteine [122]. However, MDSCs do not express the ASC transporter required for cysteine export, thus depriving T cells of cysteine. Of note, neither T cells nor MDSCs can synthesize cystathionine due to the lack of cystathionase, an enzyme that allows for conversion of intracellular methionine into cysteine [107]. The deprivation of cysteine in T lymphocytes induces metabolic modifications that lead to alteration of their functions and higher oxidative stress [122]. Altogether, the immunosuppressive microenvironment built up by MDSCs fosters the metabolic competition for cysteine by sequestering cystine, which enhances the inhibition of T cells responses and leads to their exhaustion [122]. 

Only a limited number of clinical trials have evaluated the possible interest in targeting cysteine metabolism. For instance, the role of cysteine supplementation through N-acetyl-cysteine (NAC) oral administration and its effects on treatment responses is currently being assessed without any published results (Figure 2). In mouse models, evidence has demonstrated an increased control of tumor growth in vivo in a melanoma mouse model [123] and an increased efficacy of adoptive T cell transfer [124]. In an in vivo breast cancer model, NAC administration was associated with a decrease in immunosuppressive MDSCs differentiated into non-suppressive macrophages [122]. Currently, the effect of NAC has mostly been assessed as a detoxication agent in preventing chemotherapy toxicities. The main issue of this strategy is that the absorption of NAC after oral intake may be uncertain, and its real effect on antitumor therapy efficacy and their immunomodulatory effects could be difficult to assess. 

A promising clinical target is the xc- antiporter. Cystine enters MDSCs through this antiporter and is then metabolized into cysteine and glutathione, an important factor in treatment resistance, mainly through detoxification of the cells [125]. Indeed, the expression of the xc- antiporter has a negative impact on cancer outcomes, as it is associated with increased glutathione production and chemotherapy resistance [126]. Furthermore, the expression of the xc- antiporter decreases levels of ROS, leading to enhanced cancer cell survival [127]. It has been shown in mice that the indirect inhibition of this antiporter could sensitize cisplatin-resistant cancer cells and increase their cell death upon treatment [128]. As it is expressed by myeloid and tumor cells, the inhibition of xc- antiporter could be an interesting therapeutic option. However, due to its importance in APC and T cell survival and proliferation, the systemic inhibition of xc- antiporter could be challenging. Sulfasalazine can block this transporter, leading to decreased intracellular concentrations and increased extracellular pool of cystine (Figure 2). Sulfasalazine is currently available in ulcerative colitis and has demonstrated in vitro an efficacy in restoring the sensitivity of tumor cells to chemotherapies [129]. However, the direct impact of this molecule on myeloid cells and on restoring antitumor immune responses in humans has yet to be determined.

As stressed here, there have been few clinical trials that have assessed the potential interest in cysteine targeting in cancer, particularly as it relates to the modulation of the functions of MDSCs and TAMs. One of the current limitations in the field is the absence of a specific drug, which is usable in clinical trials, that interferes with cysteine metabolism. In addition, considering the primary role of cysteine, not only in malignant cells but also in the control of immune function, combinatorial approaches associating cysteine-targeting strategies to augment antitumor immunity and immune based therapies deserve further investigations for patients with malignancies. In this context, a comprehensive examination of the impact of interfering with cysteine metabolism on immunosuppressive cells like TAMs and MDSCs, and more broadly on antitumoral immunity, is needed.

## 5. Extracellular Adenosine

Adenosine is related to ATP, which can be released into the microenvironment under several conditions such as hypoxia and inflammation after cell death, and their presence in an environment is sensed by purinergic receptors. ATP can be rapidly hydrolyzed into adenosine by several enzymes, such as CD39 (ectonucleoside triphosphate diphosphohydrolase-1 (ENTPD1)), which converts ATP or ADP into AMP, and CD73 (ecto-5′-nucleotidase (NT5E)), which then hydrolyzes AMP into adenosine. Once in the extracellular microenvironment, adenosine can be degraded by adenosine deaminase (ADA) [130]. However, in the context of cancer, due to the increased release of ATP and the overexpression of CD39 and/or CD73, accumulating extracellular adenosine interacts with the P1 family of receptors, including A2AR, leading to increases in intracellular levels of cAMP. In turn, cAMP induces the secretion of TGF-β, IL-10 and the expression of inhibitory immune checkpoints while decreasing the production of proinflammatory cytokines such as IFN-γ and TNF-α.

Purinergic receptor activation by ATP and adenosine is involved in multiple tumor processes, such as invasion/migration through EMT activation, proliferation, and angiogenesis [131]. As it relates to the immune system, adenosine is involved in immunosuppression and has been associated with reduced immune infiltration of tumors [131]. Indeed, adenosine receptors are also expressed at the surface of lymphoid cells, and their activation leads to the inhibition of CD4^+^ and CD8^+^ T lymphocytes, thus impairing their ability to secrete proinflammatory cytokines and reprogramming them towards a regulatory phenotype [132,133]. Furthermore, adenosine also impairs NK cell proliferation and functions. In myeloid cells, the activation of the adenosine pathway results in the recruitment of MDSCs and promotes their secretion of proangiogenic molecules [134]. Furthermore, adenosine triggers the accumulation of TAMs with dampened antitumor functions and the activation of tumor-promoting M2 macrophages [134]. It has additionally been shown that adenosine metabolism is involved in acquired resistance to ICI and chemotherapies in human cancer [135]. Furthermore, adenosine secreted by tumor cells and by MDSCs in the TME has been reported to act through an autocrine loop, further promoting MDSCs. 

Different strategies have been developed to target adenosine, such as the inhibition of CD39 or CD73 to decrease adenosine extracellular levels, the blockade of the interaction between adenosine and its receptor A2AR, and the direct inhibition of the receptor. Several molecules have thus been generated, with some of them being bispecific antibodies targeting receptors and ectonucleotidases. For instance, oleclumab, an anti-CD73 monoclonal antibody, showed promising results in stage III NSCLC treated by chemoradiotherapy followed by durvalumab (anti-PD-L1) maintenance over 1 year in the COAST trial (Figure 1). The median follow-up remained short, but the treatment with oleclumab was associated with increased ORRs and with good safety profiles [136]. These results have to be confirmed after a longer follow-up, as patients with stage III disease have prolonged survivals. Oleclumab has also been combined with durvalumab in different types of cancers (pancreas, colorectal, and NSCLC) with interesting results [137]. An increase in CD8+ T lymphocytes, granzyme-B, and PD-L1 expression in five of the six evaluable patients was observed. Of note, data related to the association of the response to oleclumab and baseline CD73 expression in this trial have yet to be published, but they will be of particular interest, as this information may provide a new tool for the better selection of patients. Targeting the CD39 enzyme could also be an interesting strategy, but no published data are yet available on the safety and efficacy of these molecules. 

Another adenosine-targeting strategy relies on antagonizing the adenosine receptor. AZD4635 was assessed as monotherapy or combined with durvalumab in advanced solid tumors (Figure 1). Results from a highly pretreated metastatic prostate cancer patient cohort were reported at ASCO 2020 [138]. The molecule was associated with a good safety profile but modest efficacy, with an ORR of 6.1% as monotherapy and 16.2% in combination. In this cohort, patients harboring a high blood adenosine gene expression signature had higher response and survival rates. Another molecule was assessed, ciforadenant, alone or combined with the anti-PD-L1 antibody atezolizumab. Again, a modest efficacy was observed in clear cell renal carcinoma [139]. Ancillary analysis showed that ciforadenant durable responses were associated with increased CD8+ T cell infiltration and correlated with the identification of an adenosine-related gene signature. One issue of these therapies is that some agents do not cross the blood–brain barrier, which could be a strong issue in several cancers that frequently metastasize to the brain. 

Therefore, targeting the adenosine receptor could be an interesting strategy, but available data from clinical trials have demonstrated limited efficacy in advanced refractory cancers. These first results in early trials further underline the need to identify predictive biomarker(s) for tumor response to better stratify patients. In this context, the biomarker may consist of the expression of the adenosine receptor within tumors or circulating adenosine levels. The use of this biomarker has to be rapid and feasible in clinical routine. Furthermore, combinations of adenosine targeting agents and/or with immunotherapies have to be developed.

## 6. The Oxidative Stress

ROS are chemically reactive molecules derived from oxygen, involved under normal conditions in many biological processes and maintained at low levels because they are toxic at high concentrations. They are produced through the mitochondrial respiratory chain from ATP metabolism or through enzymatic production by NADPH oxidase (NOX), for example. When ROS levels are significantly increased due to excessive production and/or defects in antioxidant protective processes, oxidative stress occurs. In cancer, high levels of ROS have been detected [140]. ROS are mainly produced by tumor cells, MDSCs, and Treg. In MDSCs, these molecular species are generated through NOX2 activation, and they are involved in maintaining the immature state of these cells, blocking their differentiation into macrophages [141]. Extracellular ROS also foster the recruitment of monocytes into the TME and the differentiation of TAMs into tumor-promoting M2 macrophages [142]. ROS produced by immunosuppressive cells reduce the recruitment, activation, proliferation, costimulatory receptor expression, and antitumor cytokine production of T cells [141]. Furthermore, ROS also induce NK cell apoptosis and dysfunction [143]. 

Different antioxidant molecules have been developed. CDDO-Me is a synthetic triterpenoid that induces the upregulation of detoxicating agents such as NAD(P)H, the which targets ROS-producing MDSCs [144] (Figure 1). This molecule is currently being assessed in different types of cancer, but no report is yet available regarding its effects. ATRA (all-trans retinoic acid) is an active metabolite of vitamin A that promotes the production of glutathione, which is involved in ROS detoxication [145] (Figure 2). This molecule, used in hematology, is currently being evaluated in different cancers, but its impact on myeloid cells remains to be delineated. The combination of ATRA with ipilimumab, an anti-CTLA4 antibody, was explored in melanoma patients (NCT02403778) [146]. No safety negative signal was reported, and ATRA was associated with a decreased circulation of MDSCs and increased circulation of mature myeloid cells. 

Only one molecule has been developed as a NOX2 inhibitor, histamine dihydrochloride [147] (Figure 1). In a phase II trial including patients with metastatic colorectal cancer, the efficacy of this molecule was limited [148]. The future of the development of ROS therapy could be its combination with immunotherapy, as preclinical in vitro studies have reported a synergistic effect of ROS produced after cytotoxic T cell activation and anti-PD-1 therapy [149].

However, as for all the metabolic pathways described in the previous section, strategies aimed at interfering with oxygen species production are limited by the ubiquitous involvement of ROS in many physiological processes. Indeed, ROS represent essential molecules responsible for M1-mediated cytotoxicity, so inhibiting ROS may have equivocal effects on both the tumor-promoting and tumor-suppressing effects of M2. Determining the balance of efficient ROS inhibition that would impair their immunosuppressive impact without interfering with their antitumor function thus remains a major challenge in the field. An additional question is related to the lack of predictive biomarkers and techniques allowing for the dosage of ROS into the TME.

## 7. Conclusions

Although the metabolic pathways engaged in cancer cells have been extensively studied for years, which has prompted the development of many different targeting therapies, the modification of the metabolism of immune cells and the dedicated metabolic specificities of tumor-promoting versus tumor-suppressing immune cells have received less attention. The possibility of modulating metabolic pathways in immune cells may, however, open new promising therapeutic possibilities, particularly as it relates to protumoral myeloid cells involved in cancer immune escape and resistance to therapies. Particularly, the metabolic reprogramming of immunosuppressive cells, such MDSCs and TAMs, may be of considerable interest to improve the efficacy of already established treatments such as immune checkpoint inhibitors and other types of immunotherapies. As we outlined in the current review, it is noteworthy that, in most cases, clinical trials have focused on T lymphocytes and rarely evaluated metabolic parameters in myeloid cells. More importantly, there is a current need to develop clinically usable agents that can selectively interfere with specific metabolic cascades in tumor-promoting myeloid cells due to their strong involvement in therapy resistance and tumor aggressiveness. An important issue in the development of such strategies is that metabolic pathways are strongly involved in several biological processes, and in vitro and in vivo efficacy could be translated into severe even fatal toxicities in humans. Likewise, the identification of biomarkers that may predict responses to metabolic modulators, which would significantly help guide therapeutic decisions and better select patients and to limit the risk of adverse events that could impair patients’ outcome, is still awaited. In humans, the identification of such biomarkers is challenging, as access to tumor tissue is not as easy as in animal models and surrogate biomarkers, such as circulating cells and metabolites, have to be defined. However, these surrogate biomarkers may not represent the complex interactions between tumor and immune cells within the tumor microenvironment. 

As discussed here, the targeting of specific metabolic pathways in some cells but not others is a primary challenge in the field of oncology, which further supports for the critical need to better understand the mechanisms underlying the dedicated metabolic pathways triggered in cancer versus immune cells and their interaction with conventional therapies. Negative data from clinical trials, such as the Echo 301 designed to evaluate the clinical potential of IDO1 inhibitors, underline the requirement for a more accurate designs of clinical trials and the identification of better specific biomarkers that would contribute to the better and efficient stratification of responder versus non-responders. Moreover, most clinical trials have focused on targeting the metabolic contexture of the tumor microenvironment to potentiate the response to chemotherapy or to immune checkpoints inhibitors. Such combinatorial approaches should now be more widely deployed in current immune-based therapies such as those with ICI or CAR T cells, as well as with chemotherapies that also exert immunomodulatory functions. A major concern of such combined therapies is the occurrence of toxicities that could be severe, with murine models that may not be the most relevant. This is very important because some patients will be treated under the curative adjuvant intent, and efficacy and patient outcomes in this setting are correlated to compliance to adjuvant therapies, which is strongly related to tolerance.

Besides the metabolic pathways described in the current review, the possible therapeutic interest of manipulating other metabolic cascades, such as the mitochondria and respiratory chain or lipid metabolism, should be considered.

Finally, because the metabolic pathways engaged during tumor development are highly dependent on the cancer type and may vary on a single cell basis, their targeting has to be adapted and optimized in a “personalized” manner. Indeed, the inhibition of a given metabolic pathway (i.e., glycolysis) may in theory have different effects on myeloid cells depending on the type of cancer (e.g., in lung cancer, glycolysis promotes M2-like TAM polarization mediated by HIF1α, while in breast cancer, it promotes M2-like TAM polarization through activation by lactates). All of this information highlights the need to develop personalized and adaptable metabolic immunomodulatory therapies for patients.

## Figures and Tables

**Figure 1 cells-11-00554-f001:**
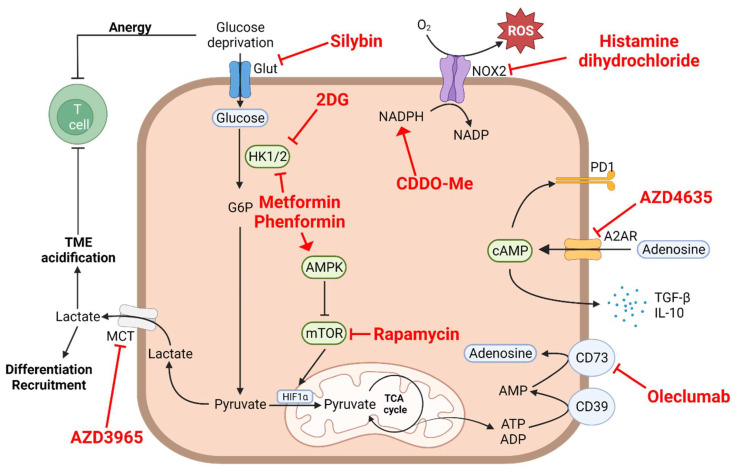
Myeloid cell intrinsic metabolism. Clinical targeting of a specific metabolic pathway regulating immunosuppressive myeloid cell functions is very likely to impact other metabolic pathways essential for their protumoral activity. Drugs currently used in clinical trials targeting different metabolic pathways are depicted in this figure in red. 2DG, 2-deoxyglucose; A2AR, adenosine A2A receptor; ADP, adenosine diphosphate; AMP, adenosine monophosphate; AMPK, 5′ adenosine monophosphate-activated protein kinase; ATP, adenosine triphosphate ATRA, all-trans retinoic acid; cAMP, cyclic adenosine monophosphate; CDDO-Me, bardoxolone methyl; G6P, glucose-6-phosphate; Glut, glucose transporter; HIF1α, hypoxia-inducible factor 1-alpha; HK1/2, hexokinase 1/2; IL-10, interleukin 10; MCT, monocarboxylate transporter; NOX2, NADPH oxidase 2; O_2_, dioxygen; PD1, programmed cell death protein 1; ROS, reactive oxygen species; TCA, tricarboxylic cycle; TGF-β, transforming growth factor β; TME, tumor microenvironment.

**Figure 2 cells-11-00554-f002:**
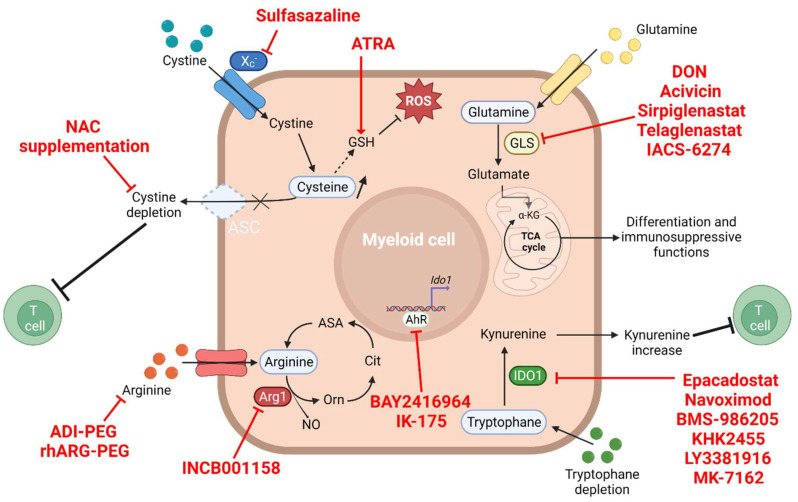
Clinical targeting of amino acid metabolism in immunosuppressive myeloid cells. Glutamate, cysteine, tryptophane, and arginine are the principal amino acids responsible for the immunosuppressive phenotypes of myeloid cells such as MDSCs and TAMs. Drugs currently used in clinical trials targeting different metabolic pathways are depicted in this figure in red. The dotted line represent multiple metabolic steps that are not detailed. ADI-PEG, pegylated arginine deiminase; AhR, aryl hydrocarbon receptor; ASA, argininosuccinate; ASC, alanine–serine–cysteine transporter; ATRA, all-trans retinoic acid; Cit, citrulline; DON, 6-diazo-5-oxo-L-norleucine; GLS, glutaminase; GSH, glutathione; IDO1, indoleamine 2,3 dioxygenase-1; NAC, N-acetyl-cysteine; NO, nitric oxide; Orn, ornithine; rhARG-PEG, pegylated recombinant human arginase; ROS, reactive oxygen species; TCA cycle, tricarboxylic acid cycle; X_C_^−^, cystine-glutamate antiporter.

**Table 1 cells-11-00554-t001:** Data of clinical trials assessing molecules that target metabolism in cancers.

Aerobic Glycolysis, Lactate Accumulation and Acidification of the Medium					
Target	Drugs	Phases	Tumor Types	NCT Number	Enrollment
GLUT4	Silybin	I	Advanced hepatocellular carcinoma	NCT01129570	3
		II	Prostate cancer	NCT00487721	12
HK2	2DG	I	Lung, breast, pancreatic, head and neck, gastric cancer	NCT00096707	50
		I/II	Advanced prostate cancer	NCT00633087	12
HK1/2	Metformin	II	Non-small cell lung cancer (NSCLC)	NCT03048500	17
		II	Small cell lung cancer (SCLC)	NCT03800602	24
		II	Small cell lung cancer (SCLC)	NCT03994744	68
		II	Solid tumor	NCT04114136	108
MCT1	AZD3965	I	Adult solid tumor | Diffuse large B cell lymphoma | Burkitt lymphoma	NCT01791595	53
mTOR	rapamycin				
**Glutamine and Glutaminolysis**					
**Target**	**Drugs**	**Phases**	**Tumor Types**	**NCT Number**	**Enrollment**
Glutamine	DRP-104 (Sirpiglenastat)	I/II	Solid tumor	NCT04471415	246
Glutaminase (GLS)	Teleglenastat (CD-839)	I	Ovarian cancer	NCT03944902	33
		I	Multiple myeloma	NCT03798678	36
		I	Non-small cell lung cancer (NSCLC)	NCT03831932	18
		I	Non-small cell lung cancer (NSCLC)	NCT04250545	85
		I	Hematological tumors	NCT02071888	25
		I	Solid tumor	NCT02071862	210
		I	Anaplastic or diffuse astrocytoma	NCT03528642	40
		I	Acute myeloid leukemia (AML)|Acute lymphocytic leukemia (ALL)	NCT02071927	43
		I/II	Advanced myelodysplastic syndrome	NCT03047993	40
		I/II	Colorectal cancer	NCT03263429	40
		I/II	Clear cell renal cell carcinoma (ccRCC)|Melanoma|Non-small cell lung cancer (NSCLC)	NCT02771626	118
		I/II	Solid tumor	NCT03875313	33
		I/II	Solid tumor	NCT03965845	53
		II	Triple-negative breast cancer (TNBC)	NCT03057600	52
		II	Prostate cancer metastatic	NCT04824937	30
		II	Solid tumor	NCT03872427	108
		II	Non-squamous non-small cell lung cancer (NSCLC)	NCT04265534	120
		II	Renal cell carcinoma	NCT03428217	445
		II	Renal cell carcinoma	NCT03163667	63
	IACS-6274	I	Solid tumor	NCT05039801	36
**Arginine**					
**Target**	**Drugs**	**Phases**	**Tumor Types**	**NCT Number**	**Enrollment**
Arginase 1/2 inhibitors	INCB001158	I	Solid tumor	NCT03910530	18
		I/II	Solid tumor	NCT03361228	5
		I/II	Solid tumor	NCT03314935	149
		I/II	Solid tumor	NCT02903914	260
		I/II	Multiple myeloma	NCT03837509	12
Arginine depletion	ADI-PEG	I	HER2-negative breast cancer	NCT01948843	15
		I	Solid tumors|Prostate cancer	NCT01497925	43
		I	Advanced pancreatic cancer	NCT02101580	21
		I	Advanced solid cancers	NCT03254732	47
		I	Solid tumor	NCT01665183	8
		I	Solid tumor	NCT02029690	85
		I	Metastatic melanoma	NCT00029900	15
		I	Hepatocellular carcinoma	NCT02101593	8
		I	Uveal melanoma	NCT03922880	9
		I	Acute myeloid leukemia	NCT05001828	60
		I	Acute myeloid leukemia	NCT02875093	23
		I	Glioblastoma	NCT04587830	32
		I/II	Advanced gastrointestinal (GI) malignancies|Hepatocellular carcinoma|Gastric cancer|Colorectal cancer	NCT02102022	140
		I/II	Metastatic melanoma|Skin cancer|Neoplasm	NCT00520299	31
		II	Hepatocellular carcinoma	NCT00056992	34
		II	Melanoma (skin)	NCT00450372	38
		II	Soft tissue sarcoma	NCT03449901	98
		II	Re-sectable hepatocellular carcinoma	NCT04965714	10
		II	Acute myeloid leukemia	NCT01910012	43
		II	Small cell lung cancer	NCT01266018	22
		II	Hepatocellular carcinoma	NCT02006030	30
		II	Non-Hodgkin’s lymphoma	NCT01910025	18
		II/III	Mesothelioma	NCT02709512	386
		III	Hepatocellular carcinoma	NCT01287585	636
**Tryptophane Kynurenine**					
**Target**	**Drugs**	**Phases**	**Tumor Types**	**NCT Number**	**Enrollment**
IDO inhibitors	Epacadostat (INCB024360)	I	Rectal cancer	NCT03516708	39
		I	Solid tumors and hematologic malignancy	NCT01195311	52
		I	Ovarian cancer|Fallopian tube carcinoma|Primary peritoneal carcinoma	NCT02118285	2
		I	NSCLC (non-small cell lung carcinoma)|UC (urothelial cancer)	NCT02298153	29
		I	Advanced solid tumor|Non-small cell lung cancer (NSCLC)	NCT03217669	22
		I	Solid tumors	NCT02559492	142
		I	Unresectable or Metastatic Solid Tumors	NCT03589651	83
		I	Neoplasms|Non-small-cell lung carcinoma	NCT02862457	34
		I	Glioblastoma|Glioblastoma multiforme	NCT03707457	3
		I	Solid tumor	NCT03471286	2
		I	Fallopian and ovarian cancer	NCT02042430	17
		I/II	Melanoma	NCT01604889	136
		I/II	Fallopian and ovarian cancer	NCT02166905	40
		I/II	Advanced solid tumors|Lymphoma	NCT03322384	20
		I/II	Solid tumors	NCT02178722	444
		I/II	B-cell malignancies|Colorectal cancer (CRC)|Head and neck cancer|Lung cancer|Lymphoma|Melanoma|Ovarian cancer|Glioblastoma	NCT02327078	307
		I/II	Epithelial ovarian cancer|Tube cancer|Peritoneal cancer	NCT02785250	85
		I/II	Solid tumor	NCT03361228	5
		I/II	Solid tumor	NCT02959437	70
		I/II	Breast cancer	NCT03328026	60
		I/II	Platinum-resistant ovarian cancer|Platinum-resistant fallopian cancer|Platinum-resistant peritoneal cancer	NCT02575807	35
		I/II	Solid tumor	NCT03085914	70
		I/II	Solid tumors|Head and neck cancer|Lung cancer|UC (urothelial cancer)	NCT02318277	176
		I/II	Solid tumor	NCT03347123	11
		I/II	Solid tumors	NCT03277352	10
		II	Gastrointestinal stromal tumors	NCT03291054	1
		II	Metastatic pancreatic adenocarcinoma	NCT03006302	44
		II	Ovarian cancer|Genitourinary (GU) tumors	NCT01685255	83
		II	Lung cancer	NCT03322566	233
		II	Head and neck carcinoma	NCT03463161	2
		II	Lung cancer	NCT03322540	154
		II	Colorectal cancer	NCT03196232	3
		II	Thymic carcinoma|Thymus neoplasms|Thymus cancer	NCT02364076	45
		II	Head and neck carcinoma	NCT03823131	14
		II	Melanoma	NCT01961115	11
		II	Endometrial cancer	NCT04463771	220
		II	Malignant ovarian clear cell tumor|Recurrent ovarian carcinoma	NCT03602586	14
		II	Sarcoma	NCT03414229	30
		II	Urothelial carcinoma	NCT04586244	45
		II	Glioma|Glioblastoma	NCT03532295	55
		II	Myelodysplastic syndromes	NCT01822691	15
		III	Lung cancer	NCT03348904	2
		III	UC (urothelial cancer)	NCT03361865	93
		III	UC (urothelial cancer)	NCT03374488	84
		III	Head and neck cancer	NCT03358472	89
		III	Renal cell carcinoma (RCC)	NCT03260894	129
		III	Melanoma	NCT02752074	706
	Linrodostat (BMS-986205)	II	Bladder cancer|Bladder tumors|Bladder neoplasms	NCT03519256	69
		I/II	Advanced cancer	NCT03792750	17
		I	Advanced cancer	NCT03192943	11
		III	Bladder cancer|Muscle-invasive bladder cancer|BMS-986205	NCT03661320	1200
		I/II	Advanced cancer|Melanoma|Non-small cell lung cancer	NCT02658890	630
		II	Endometrial adenocarcinoma|Endometrial carcinosarcoma	NCT04106414	50
		I/II	Advanced cancer	NCT03459222	184
		III	Melanoma|Skin cancer	NCT03329846	20
		I	Multiple malignancies	NCT03346837	53
		I	Cancer	NCT03247283	9
		I/II	Hepatocellular carcinoma	NCT03695250	8
		II	Advanced cancer	NCT02750514	295
		II	Head and neck carcinoma	NCT03854032	48
		II	Solid tumors	NCT02996110	200
		II	Advanced gastric cancer	NCT02935634	186
		I	Solid tumors	NCT03335540	50
		I	Glioblastoma	NCT04047706	30
	KHK2455	I	Solid tumors	NCT02867007	36
		I	Urothelial carcinoma	NCT03915405	50
	LY3381916	I	Solid tumors	NCT03343613	60
AhR inhibitors	BAY 2416964	I	Solid tumors	NCT04999202	78
		I	Solid tumors	NCT04069026	141
	IK-175	I	Solid tumors, urothelial carcinoma	NCT04200963	93
**Cysteine**					
**Target**	**Drugs**	**Phases**	**Tumor Types**	**NCT Number**	**Enrollment**
Cysteine supplementation	N-acetylcysteine (NAC)	I	Breast cancer	NCT01878695	13
		I	Brain tumors	NCT00238173	2
		I	Lymphoma	NCT05081479	32
		I/II	Peritoneal cancer|Mucinous adenocarcinoma	NCT03976973	100
		I/II	Ovarian cancer	NCT04520139	102
		II	Ovarian cancer	NCT02569957	1
		II	Melanoma	NCT00003346	80
Xc- antiporter	Sulfasalazine	II	Breast cancer	NCT03847311	40
**Adenosine**					
**Target**	**Drugs**	**Phases**	**Tumor Types**	**NCT Number**	**Enrollment**
CD73 inhibitors	Oleclumab	I	Bladder cancer	NCT03773666	24
		I	Solid tumors	NCT04261075	57
		I	Solid tumors	NCT03736473	6
		I	Solid tumors	NCT02503774	192
		I	Non-small cell lung cancer (NSCLC)	NCT03819465	212
		I/II	Triple-negative breast cancer (TNBC)	NCT03742102	203
		I/II	Triple-negative breast cancer (TNBC)	NCT03616886	129
		I/II	Pancreatic adenocarcinoma	NCT03611556	208
		I/II	Colorectal cancer	NCT04068610	60
		II	Non-small cell lung cancer (NSCLC)	NCT05061550	140
		II	Breast cancer luminal B	NCT03875573	147
		II	Pancreatic cancer	NCT04940286	30
		II	Non-small cell lung cancer (NSCLC)	NCT03822351	189
		II	Non-small cell lung cancer (NSCLC)	NCT03794544	84
		II	Non-small cell lung cancer (NSCLC)	NCT03334617	420
		II	Non-small cell lung cancer (NSCLC)	NCT03833440	120
		II	Sarcoma	NCT04668300	75
	AZD3965	I	Solid tumors	NCT03980821	10
		II	Prostate cancer	NCT04495179	30
	Oleclumab, AZD4635	I	Solid tumors, non-small cell lung cancer (NSCLC), prostate cancer, colorectal cancer	NCT02740985	313
		I/II	Non-small cell lung cancer (NSCLC)	NCT03381274	43
		II	Prostate cancer	NCT04089553	59
**Oxidative Stress**					
**Target**	**Drugs**	**Phases**	**Tumor Types**	**NCT Number**	**Enrollment**
NAD(P)H	CDDO-Me	I	Advanced solid tumors|Lymphoid malignancies	NCT00529438	47
		I	Lymphoid malignancies|Solid tumors	NCT00508807	21
		I/II	Pancreatic neoplasms|Pancreatic cancer	NCT00529113	33
Glutathione	ATRA	II	Advanced melanoma	NCT02403778	10
NOX2	Histamine dihydrochloride	II	Colorectal neoplasms	NCT01722162	47
		III	Leukemia	NCT00003991	360
		IV	Acute myeloid leukemia	NCT01347996	84
			Acute myeloid leukemia	NCT01770158	8

## Data Availability

Not applicable.

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
