# Peer review of "Influence of the Metabolism on Myeloid Cell Functions in Cancers: Clinical Perspectives"

_cells, 2022, doi:10.3390/cells11030554_

Round 1
Reviewer 1 Report
This is a very well done and very comprehensive review of metabolic pathways that are used by malignant cells to induce immunosuppressive tumor microenvironment and attenuate function of cytotoxic T cells, including a summary of therapeutic strategies that target these processes and are currently in pre-clinical or clinical development. The review is well organized. The figures are helpful although some abbreviations are not explained. There are minor English grammar and style errors which I am sure will be corrected during the editing process by the journal. Below are a few minor comments that I hope will be constructive and help improve the manuscript.
Comment 1:
A table summarizing ongoing clinical trials testing all the agents targeting metabolic pathways in cancer in order to reverse immunosuppressive tumor microenvironment that are discussed in the manuscript would add great value to this review and make it a useful tool for clinical and translational investigators. I suggest searching for ongoing studies at www.clinicaltrials.gov website.
Comment 2:
The following sentence on page 2 (section 2, starting in line 68) looks incomplete.
“Indeed, accumulating evidence has demonstrated that these cells can directly promote primary tumor cell survival and proliferation 69 by fostering tumor neoangiogenesis and [8].”
The next sentence also looks incomplete:
“They are also capable of remodeling the extracellular matrix and participate in local tissue invasion by [9].”
Comment 3:
Sentence in section 2 on page 5 (line 220):
“In human, mTOR inhibition has not shown significant interest and is clinically challenging insofar as mTOR is critically involved in multiple physiological processes such as the control of immune cell differentiation and functions.”
Everolimus is one mTOR inhibitor which has been approved and has significant anti-tumor activity multiple cancers, including breast cancer, renal cell carcinoma, neuroendocrine carcinoma and subependymal giant cell astrocytoma. The statement above is therefore inaccurate and should be edited.
Comment 4:
Sentence in section 3.2 on page 9 (starting in line 419):
“Regarding the results of the pivotal chemotherapy trial in the same clinical setting, INCB001158 efficacy appears modest in the first-line setting.”
Please indicate what cancer type was enrolling to that trial and whether the “first-line setting” refers to early stage or metastatic disease.
Author Response
This is a very well done and very comprehensive review of metabolic pathways that are used by malignant cells to induce immunosuppressive tumor microenvironment and attenuate function of cytotoxic T cells, including a summary of therapeutic strategies that target these processes and are currently in pre-clinical or clinical development. The review is well organized. The figures are helpful although some abbreviations are not explained.
We thank Reviewer #1 for her/his thorough proofreading and for her/his helpful comments. Abbreviations have been explicated in the revised manuscript.
There are minor English grammar and style errors which I am sure will be corrected during the editing process by the journal.
The entire manuscript has been thoroughly reviewed for grammatical and other errors.
Below are a few minor comments that I hope will be constructive and help improve the manuscript.
Comment 1:
A table summarizing ongoing clinical trials testing all the agents targeting metabolic pathways in cancer in order to reverse immunosuppressive tumor microenvironment that are discussed in the manuscript would add great value to this review and make it a useful tool for clinical and translational investigators. I suggest searching for ongoing studies at www.clinicaltrials.gov website.
We thank Reviewer #1 for her/his comment. As suggested, we have added Table 2 that summarizes ongoing clinical trials for molecules listed in our review.
Comment 2:
The following sentence on page 2 (section 2, starting in line 68) looks incomplete.
“Indeed, accumulating evidence has demonstrated that these cells can directly promote primary tumor cell survival and proliferation 69 by fostering tumor neoangiogenesis and [8].”
The next sentence also looks incomplete:
“They are also capable of remodeling the extracellular matrix and participate in local tissue invasion by [9].”
We thank the reviewer for this comment. We have modified the first sentence in the revised manuscript, and we have removed the word “by” in the second sentence, page 2 lines 70 and 71.
Comment 3:
Sentence in section 2 on page 5 (line 220):
“In human, mTOR inhibition has not shown significant interest and is clinically challenging insofar as mTOR is critically involved in multiple physiological processes such as the control of immune cell differentiation and functions.”
Everolimus is one mTOR inhibitor which has been approved and has significant anti-tumor activity multiple cancers, including breast cancer, renal cell carcinoma, neuroendocrine carcinoma and subependymal giant cell astrocytoma. The statement above is therefore inaccurate and should be edited.
We agree with Reviewer #1 that the sentence was unclear. We wanted to underline that metabolic targeting through mTOR inhibition has not been specifically addressed in clinical trials using mTOR inhibitors. We have modified the manuscript, page 5 lines 225-226.
Comment 4:
Sentence in section 3.2 on page 9 (starting in line 419):
“Regarding the results of the pivotal chemotherapy trial in the same clinical setting, INCB001158 efficacy appears modest in the first-line setting.”
Please indicate what cancer type was enrolling to that trial and whether the “first-line setting” refers to early stage or metastatic disease.
We have added the type and stage of cancer disease involved in this trial, page 10 line 443. Indeed, it was advanced or metastatic biliary tract cancers, meaning that the first-line setting referred to metastatic disease.
Reviewer 2 Report
This is well written comprehensive review that will be useful for readers interested in the field of metabolism-associated features of tumor-infiltrating myeloid cells with immunosuppressive characteristics such as M2-macrophages and myeloid-derived suppressor cells (MDSCs).
There are several specific issues that need to be addressed.
Introduction. I invite the authors to avoid the citation of manuscript in preparation considering that the concept about the impact of macrophages/MDSCs are able to sustain cancer by no-immune mechanisms has been already demonstrated. I suggest the authors to use key references such as Ugel S et al. Tumor-induced myeloid deviation: when myeloid-derived suppressor cells meet tumor-associated macrophages. J Clin Invest. 2015; Kitamura T et al. Immune cell promotion of metastasis. Nat Rev Immunol. 2015; Mantovani A, et al. Tumour-associated macrophages as treatment targets in oncology. Nat Rev Clin Oncol. 2017 and Veglia F, et al. Myeloid-derived suppressor cells in the era of increasing myeloid cell diversity. Nat Rev Immunol. 2021.- Chapter 1. The authors summarize MDSC and macrophages biological features shortly. To avoid possible misunderstanding in the reader, who has a not solid background in the field of myeloid cells with immunosuppressive features, I invite the authors to insert a table in which summarize the main features of these cell subsets. Similarly, to the previous point, it is quite ambitious to use a citation of a manuscript in preparation to define MDSC in cancer and other pathological disorders. In my opinion seminal works of Gabrilovich and Bronte, who are the pioneers of MDSC field, need to be cited: Bronte et al. Recommendations for myeloid-derived suppressor cell nomenclature and characterization standards. Nat Commun. 2016; Veglia F, at al. Myeloid-derived suppressor cells coming of age. Nat Immunol. 2018. Moreover some seminal manuscripts of Mantovani and Pollard laboratory need to be cited in the macrophage description.
Finally, the authors should comment some seminal work about macrophage-related atlas obtained by scRNA-sq and the ability of new technologies to better solve cell plasticity.
- Chapter 2. General control nonderepressible 2 (GCN2), which is an environmental sensor controlling transcription and translation in response to nutrient availability, is recently described to control macrophages and MDSCs functions.
- Chapter 3.2 “Arginine depletion”. As correctly described by the authors, arginine and arginases (ARG1 and ARG2) impact on tumor microenvironment since the depletion of arginine promotes T cell inhibition. In light of these premises, the authors should cite the excellent work of Federica Sallusto and Roger Geiger about the ability to restore arginine using engineered bacteria (Canale F et al. Metabolic modulation of tumours with engineered bacteria for immunotherapy. Nature. 2021). To block ARG1 functions some researcher identified putative transcriptional ARG1-associated inhibitor such as AT38 drug (Molon B et al. Chemokine nitration prevents intratumoral infiltration of antigen-specific T cells. J Exp Med. 2011). This drug recently has been described to alter immunosuppressive functions of both M2-macrophages and MDSCs as well as to alter the immune landscape of pancreatic cancers in preclinical setting (De Sanctis F. at al. Interrupting the nitrosative stress fuels tumor-specific cytotoxic T lymphocytes in pancreatic cancer. J Immunother Cancer. 2022). ARG1 in cooperation with iNOS activity can fuel local immunosuppression by the generation of reactive nitrogen species (RNS) that limit migration and function of tumor.-infiltrating T cells. This aspect deserves to be discussed briefly in the manuscript. Finally, a promising approach is to promote an immune response against ARG1/IDO1 by expanded specific T cells able to eliminate immunosuppressive myeloid cells as highlighted by Mads Hald Andersen group (Aaboe Jørgensen M, et al. Arginase 1-Based Immune Modulatory Vaccines Induce Anticancer Immunity and Synergize with Anti-PD-1 Checkpoint Blockade. Cancer Immunol Res. 202; Kjeldsen J et al. A phase 1/2 trial of an immune-modulatory vaccine against IDO/PD-L1 in combination with nivolumab in metastatic melanoma.Nat Med. 2021).
- Chapter 3.3 Tryptophan-kynurenin pathway. The authors should stress two major concepts about IDO1 biology: a) ARG-expressing myeloid cells (DCs) produce orninthine and spermidine, and these in turn promote the phosphorylation of IDO1 and its long-term signalling activity (Mondanalli G. et al Immunity 2017). Therefore, these two metabolic pathways are strictly correlated. b) IDO1 is not only an enzyme, which can be targetable but also it acts as a signal-transducing molecule and fine-tunes the immune response over the long term in a TGF-dependent manner. This aspect needs to be stressed in the manuscript.
Author Response
This is well written comprehensive review that will be useful for readers interested in the field of metabolism-associated features of tumor-infiltrating myeloid cells with immunosuppressive characteristics such as M2-macrophages and myeloid-derived suppressor cells (MDSCs).
There are several specific issues that need to be addressed.
- I invite the authors to avoid the citation of manuscript in preparation considering that the concept about the impact of macrophages/MDSCs are able to sustain cancer by no-immune mechanisms has been already demonstrated. I suggest the authors to use key references such as Ugel S et al. Tumor-induced myeloid deviation: when myeloid-derived suppressor cells meet tumor-associated macrophages. J Clin Invest. 2015; Kitamura T et al. Immune cell promotion of metastasis. Nat Rev Immunol. 2015; Mantovani A, et al. Tumour-associated macrophages as treatment targets in oncology. Nat Rev Clin Oncol. 2017 and Veglia F, et al. Myeloid-derived suppressor cells in the era of increasing myeloid cell diversity. Nat Rev Immunol. 2021.
We included the proposed citations in the introduction as suggested by the Reviewer #2.
- Chapter 1. The authors summarize MDSC and macrophages biological features shortly. To avoid possible misunderstanding in the reader, who has a not solid background in the field of myeloid cells with immunosuppressive features, I invite the authors to insert a table in which summarize the main features of these cell subsets.
We agree with Reviewer #2’s comment, and we have added a table summarizing the features of macrophages and MDSC (Table 1).
- Similarly, to the previous point, it is quite ambitious to use a citation of a manuscript in preparation to define MDSC in cancer and other pathological disorders. In my opinion seminal works of Gabrilovich and Bronte, who are the pioneers of MDSC field, need to be cited: Bronte et al. Recommendations for myeloid-derived suppressor cell nomenclature and characterization standards. Nat Commun. 2016; Veglia F, at al. Myeloid-derived suppressor cells coming of age. Nat Immunol. 2018. Moreover some seminal manuscripts of Mantovani and Pollard laboratory need to be cited in the macrophage description.
Finally, the authors should comment some seminal work about macrophage-related atlas obtained by scRNA-sq and the ability of new technologies to better solve cell plasticity.
We agree with Reviewer #2’s remark. Indeed, the plasticity and diversity of myeloid subtypes are important and their identification remains controversial. However, our review aims at reviewing and discussing the different ways to increase response to immunotherapies by targeting myeloid metabolism, and we already discussed the heterogeneity and the plasticity of myeloid subpopulations in paragraph 1. According to Reviewer #2’s comment, we have added some elements on the role of scRNA-Seq to better identify immune subpopulations, page 2 lines 82 to 84.
- Chapter 2. General control nonderepressible 2 (GCN2), which is an environmental sensor controlling transcription and translation in response to nutrient availability, is recently described to control macrophages and MDSCs functions.
We thank Reviewer #2 for this important comment, and we added a sentence in the paragraph 1, page 3 lines 121 to 125 as suggested.
- Chapter 3.2 “Arginine depletion”. As correctly described by the authors, arginine and arginases (ARG1 and ARG2) impact on tumor microenvironment since the depletion of arginine promotes T cell inhibition. In light of these premises, the authors should cite the excellent work of Federica Sallusto and Roger Geiger about the ability to restore arginine using engineered bacteria (Canale F et al. Metabolic modulation of tumours with engineered bacteria for immunotherapy. Nature. 2021). To block ARG1 functions some researcher identified putative transcriptional ARG1-associated inhibitor such as AT38 drug (Molon B et al. Chemokine nitration prevents intratumoral infiltration of antigen-specific T cells. J Exp Med. 2011). This drug recently has been described to alter immunosuppressive functions of both M2-macrophages and MDSCs as well as to alter the immune landscape of pancreatic cancers in preclinical setting (De Sanctis F. at al. Interrupting the nitrosative stress fuels tumor-specific cytotoxic T lymphocytes in pancreatic cancer. J Immunother Cancer. 2022).
These points have been included in the revised manuscript (page 10, lines 460 to 467).
- ARG1 in cooperation with iNOS activity can fuel local immunosuppression by the generation of reactive nitrogen species (RNS) that limit migration and function of tumor.-infiltrating T cells. This aspect deserves to be discussed briefly in the manuscript.
We agree with Reviewer #2’s remark, and we added a paragraph in the arginine section, page 9 lines 395 to 409.
- Finally, a promising approach is to promote an immune response against ARG1/IDO1 by expanded specific T cells able to eliminate immunosuppressive myeloid cells as highlighted by Mads Hald Andersen group (Aaboe Jørgensen M, et al. Arginase 1-Based Immune Modulatory Vaccines Induce Anticancer Immunity and Synergize with Anti-PD-1 Checkpoint Blockade. Cancer Immunol Res. 2021; Kjeldsen J et al. A phase 1/2 trial of an immune-modulatory vaccine against IDO/PD-L1 in combination with nivolumab in metastatic melanoma.Nat Med. 2021).
We thank the reviewer for this interesting remark. A paragraph has been added page 10 lines 454 to 459 and page lines 517 to 520.
- Chapter 3.3 Tryptophan-kynurenin pathway. The authors should stress two major concepts about IDO1 biology: a) ARG-expressing myeloid cells (DCs) produce orninthine and spermidine, and these in turn promote the phosphorylation of IDO1 and its long-term signalling activity (Mondanalli G. et al Immunity 2017). Therefore, these two metabolic pathways are strictly correlated. b) IDO1 is not only an enzyme, which can be targetable but also it acts as a signal-transducing molecule and fine-tunes the immune response over the long term in a TGF-dependent manner. This aspect needs to be stressed in the manuscript.
We thank the reviewer for this interesting remark. A paragraph has been added in the sections 3.3, page 11, lines 497 to 500 and 504 to 506.
Reviewer 3 Report
This review manuscript makes a comprehensive and nice summary of tumor metabolism on myeloid cells in tumor microenvironment and further discusses treatment strategies targeting these metabolic pathways. The figure 1 and figure 2 are informative and capitalize major influential pathways and targets on tumor myeloid cells. It is good to add a table presenting the clinical trials related to these metabolic targets that have potential for clinical implication.
Some minor comments:
- It is inappropriate to cite unpublished works at the line 56 and line 91 on page 2, etc.
- The text is incomplete when references 8 and 9 are cited.
- Some grammar issues. e.g. "participate to" at line 111 on page 3 and "describe" at line 259 on page 6.
Author Response
This review manuscript makes a comprehensive and nice summary of tumor metabolism on myeloid cells in tumor microenvironment and further discusses treatment strategies targeting these metabolic pathways. The figure 1 and figure 2 are informative and capitalize major influential pathways and targets on tumor myeloid cells. It is good to add a table presenting the clinical trials related to these metabolic targets that have potential for clinical implication.
Some minor comments:
- It is inappropriate to cite unpublished works at the line 56 and line 91 on page 2, etc.
We modified this in the manuscript.
- The text is incomplete when references 8 and 9 are cited.
We thank Reviewer #3 for this comment. We have modified the first sentence in the revised manuscript, page 2 lines 70 and 71.
- Some grammar issues. e.g. "participate to" at line 111 on page 3 and "describe" at line 259 on page 6.
The manuscript has been thoroughly reviewed for grammatical and other errors.